# Cognitive Digital Intervention for Older Patients with Parkinson’s Disease during COVID-19: A Mixed-Method Pilot Study

**DOI:** 10.3390/ijerph192214844

**Published:** 2022-11-11

**Authors:** Sara Santini, Margherita Rampioni, Vera Stara, Mirko Di Rosa, Lucia Paciaroni, Susy Paolini, Simona Fioretti, Silvia Valenza, Giovanni Renato Riccardi, Giuseppe Pelliccioni

**Affiliations:** 1Centre for Socio-Economic Research on Aging, IRCCS INRCA-National Institute of Health and Science on Aging, Via Santa Margherita 5, 60124 Ancona, Italy; 2Model of Care and New Technologies, IRCCS INRCA-National Institute of Health and Science on Aging, Via Santa Margherita 5, 60124 Ancona, Italy; 3Laboratory of Geriatric Pharmacoepidemiology, IRCCS INRCA-National Institute of Health and Science on Aging, Via Santa Margherita 5, 60124 Ancona, Italy; 4Neurology Department, IRCCS INRCA-National Institute of Health and Science on Aging, Via della Montagnola 81, 60100 Ancona, Italy; 5Clinical Unit of Physical Rehabilitation, IRCCS INRCA-National Institute of Health and Science on Aging, Via della Montagnola 81, 60100 Ancona, Italy

**Keywords:** parkinson’s disease, cognition, mild cognitive impairment, dementia, neuropsychological test, COVID-19, cognitive telerehabilitation

## Abstract

Mild cognitive impairment is frequent among people with Parkinson’s disease. Cognitive training seems effective for cognitive status and for mitigating anxiety and depression. With the COVID-19 outbreak, such therapeutic interventions were delivered online. This longitudinal mixed-method study was aimed at evaluating the effectiveness of an online cognitive treatment, carried out during COVID times and based on Parkinson’s-Adapted Cognitive Stimulation Therapy, on cognitive domains and mood of 18 older people with Parkinson’s disease. After screening, the cognitive status and mood were assessed three times by Addenbrooke’s Cognitive Examination-Revised scale and the Geriatric Depression Scale-Short Form. At the follow-up, patients were also interviewed for understanding their experience with the technology. Such treatment was effective on the participants’ cognitive functions, but not on their mood. Despite some initial problems with the technology, the online intervention was experienced as a way of not being ‘left behind’, staying in contact with others, and being safe during the lockdown. This suggests that online cognitive treatment can be adopted to integrate face-to-face interventions by increasing their efficacy, accessibility, and long-term outcomes. Suggestions for future research are given.

## 1. Introduction

Parkinson’s disease (PD) is a progressive multi-system neurodegenerative disease. The average age of diagnosis is 60 years but it has been estimated that PD increases with age, reaching a prevalence of 2.6% in people aged 85 to 89 years [1]. PD affected 6.1 million individuals globally in 2016 and up to 2.5 million in 1990, confirming that it was the fastest-growing neurological disorder in terms of prevalence, disability, and deaths, mainly due to increased life expectancy [2]. PD can impair motor functions (i.e., bradykinesia, rest tremor, and rigidity, ascribed to the loss of dopaminergic neurons, and those involving posture, balance, and gait), as well as non-motor functions i.e., sleep disorders, a spectrum of neuropsychiatric symptoms (e.g., depression, apathy, hallucinations), autonomic disorders (e.g., constipation and urinary disturbances), and cognitive impairment (e.g., involvement of executive functions, memory, and visuospatial functions) up to dementia [3].

While non-motor symptoms become increasingly prevalent with advancing disease, many of them can also antedate the first occurrence of motor signs [4]. Cognitive symptoms also represent a fairly frequent problem, which becomes evident early at the time of diagnosis and during the early stages of the disease in which it is easy to find mild cognitive impairment (MCI), defined as mild cognitive deficits that do not significantly interfere with the autonomy or social behavior of the patient and that do not amount to a diagnosis of dementia [5]. MCI is found in about 26.7% of patients at onset [6], progressing to full-blown dementia [7] and, despite possible differences in figures depending on different cognitive assessment tools used for the diagnosis [8], it is estimated that about 30–40% of PD patients suffer from MCI [9]. Some of the cognitive treatments, such as stimulation (non-specific stimulation of cognitive and social functioning), training (using standardized cognitive tasks), rehabilitation (targeting specific areas of difficulty in daily living activities) [10,11,12], and cognitive training (CT), can be effective intervention strategies to improve or at least maintain cognitive levels in patients with MCI by activating brain compensation mechanisms to tackle the physiological and pathological neurodegeneration processes [13], thus slowing down progression to dementia [14]. In particular, Cognitive Stimulation Therapy (CST) is an evidence-based intervention and cost-effective psychosocial group treatment that can improve cognition, mood, quality of life, and communication in people with mild to moderate dementia [15,16], and it can also improve several cognitive dysfunctions associated with PD [17,18]. Cognitive training and rehabilitation interventions are delivered by healthcare professionals within care facilities. However, this was the practice before the COVID-19 outbreak, which radically transformed public and private healthcare organizations around the world [19] and posed unprecedented new challenges in the care of older patients with neurodegenerative diseases, including patients with PD. In fact, especially from March to June 2020, health and social care structures suspended face-to-face therapeutic interventions and cognitive rehabilitation in order to limit the virus’ spread, to the detriment of PD patients’ cognitive functions and mental health [20]. However, more than the virus itself, it was the disruption of care services providing cognitive rehabilitation [21] and the social isolation caused by physical distancing measures imposed during the lockdown that contributed to the worsening of PD symptoms [22]. Lockdown decreased the physical activity of people with PD thus aggravating psychological distress exacerbated by pre-pandemic neuropsychiatric symptoms [23], worsened motor (decrease in activities and gait, tremors) and non-motor (mood and sleep disturbances) symptoms [24], accelerated cognitive decline, and worsened the speech functions of older people with PD more than those of people with MCI not associated with PD [25]. Thus, the latter was faced with a ‘double burden,’ as the pandemic heightened their vulnerability due to increased health risks from exposure and restricted access to care services due to heightened lockdown measures [26]. Many healthcare organizations delivered at-a-distance online cognitive rehabilitation interventions to overcome the barriers imposed by physical distancing measures and guarantee continuity of support and assistance. Although the wide bulk of literature on cognitive telerehabilitation (CTR) flourished in the last two years as a consequence of the challenges posed by the outbreak, most studies are characterized by small samples and a lack of standardized procedures, aims, and targets [27]. Thus, the efficacy of CTR services needs to be further proven by research studies. Moreover, to the best of our knowledge, there are no published data on the effectiveness of cognitive digital rehabilitation for PD patients conducted during the COVID-19 pandemic [28]. Based on these premises, this study shares the results of a cognitive digital rehabilitation service performed in March 2020 that enabled the prosecution of the rehabilitation paths already started. The aim of this longitudinal mixed-method study is to evaluate the effectiveness of an online cognitive-rehabilitative treatment on individuals with a mild cognitive disorder associated with PD and to assess if the treatment’s effects on cognitive functions and mood state are comparable to those of conventional face-to-face interventions.

## 2. Materials and Methods

### 2.1. Study Design

This mixed-method (MM) study is based on a four-wave assessment of the outcomes of a digital cognitive rehabilitation intervention that involved a convenience sample of 18 PD patients (mean age 73.1 ± 4.8 years) already included in a face-to-face cognitive rehabilitation pathway. Between January and February 2020, the people enrolled in the trial received an evaluation of their cognitive status (T0) through the QUANT assessment tools described in the following section. In October 2020, after eight months of lockdown (which in Italy began on 9 March 2020), when all interventions were suspended, they were reassessed in their cognitive functions and mood (T1). After the T1 assessment, the PD patients received an intensive digital cognitive rehabilitation treatment consisting of 14 biweekly sessions, and then they were reassessed (T2; January 2021). This assessment was followed by maintenance treatment (1 session per week for six months), which was followed by a QUANT-QUAL follow-up in July 2021 (T3). The MM study design was adopted for the twofold purpose of monitoring the cognitive functions and mood of PD patients through QUANT data and shedding light on their experience during digital cognitive rehabilitation through QUAL data providing contextual understanding. In accordance with the triangulation principle [29], open-ended questions were embedded within a larger quantitative survey. First, QUANT and QUAL data were analyzed separately, and then by similarities and contrasts. The point of integration of the two analyses was reached when the QUAL findings were added and integrated with the QUANT data to explain the possible influence of the digital intervention environment and participants’ state of mind (collected through semi-structured interviews) on the intervention’s outcomes measured through quantitative measures, i.e., cognitive functions (e.g., memory, attention, and language). Quantitative and qualitative results were then brought together in the overall interpretation. Figure 1 provides an overview of the study design.

### 2.2. The Intervention’s Methodology

The intervention was based on the Parkinson’s-Adapted Cognitive Stimulation Therapy (CST) model [17]. Some adaptations were necessary to guarantee the remote activities, for example, in the presentation method which was changed to verbal rather than manual or paper-pencil, and in the level of difficulty, which was adapted to the abilities of the subject with mild cognitive impairment. As with CST [16], 14 twice-weekly sessions were planned, followed by one session per week for a duration of six months for maintenance therapy. The treatment took place in groups of 4 patients in remote mode through the Microsoft-Teams platform. All subjects maintained the prescribed drug therapy and regular physiotherapy activity during the intervention. The intervention’s primary outcome is to evaluate the effectiveness of a cognitive-rehabilitative treatment on individuals with mild cognitive impairment associated with PD. The endpoint is measured through the Mini-Mental State Examination (MMSE) [30]. The secondary outcomes are to evaluate the effect of the treatment on cognitive domains such as memory, attention, language, and visual-spatial abilities and the impact on the thymic state. These endpoints are measured through the Addembrooke’s Cognitive Examination Battery (ACE-R) [31] and the Geriatric Depression Scale: Short Form (GDS) [32].

### 2.3. Participants’ Inclusion Criteria

PD patients were included in the study if: (a) they were 65 years of age or older; (b) had not received any cognitive training between March and October 2020; (c) could count on the assistance of a family caregiver to access and use the online platform; (d) were diagnosed with PD. This criterion was assessed through the Hoehn & Yahr’s scale: 1–3 based on the UK PD Society Brain Bank [33]; (e) were diagnosed with mild cognitive impairment associated with PD [6], by a face-to-face extensive neuropsychological assessment reported in Table 1.

Moreover, participants needed to have a primary level of education or higher and a Mini-Mental State Examination (MMSE) score dated January or February 2020. Participants were excluded if they suffered from other neurological diseases, deep brain stimulation, schizophrenia, depression, and sensory deprivations that could interfere with the treatment. The local ethics committee approved the study, and informed written consent was obtained from all subjects.

We included 12 patients with single non-amnesic type MCI, 4 with multiple non-amnesic type MCI, and 2 with multiple amnesic type MCI. The onset of cognitive symptoms was between 6 and 20 months before the diagnosis of MCI. UPDRS mean scores were: Motor 18.35 ± 5.98; UPDRS II (ADL) 13.58 ± 4.44; Total 31.94 ± 9.26. One patient was not receiving any PD treatment, five patients were under low-dose levodopa monotherapy, and the other patients were receiving levodopa combined with other antiparkinsonian treatments (eight patients also had a prescription for dopamine agonists, while four also MAO-B inhibitors). Five patients were using antihypertensive medications, and none of the participants were taking medications that may influence the cognitive evaluation.

### 2.4. QUANT Outcome Measures and Statistical Analysis

Quantitative tools aimed at measuring patients’ cognitive status and cognitive sub-domains by means of the MMSE scale [30] and the ACE-R scale [44]. These scales were chosen because they are better suited to remote administration [45,46].

Patients’ moods were measured by means of the GDS [32]. The MMSE is a brief objective assessment of cognitive functioning and a measure of changes in cognitive status including items for testing temporal and spatial orientation, immediate and delayed verbal memory, language, attention, and praxis. Its total score is 30 points. The ACE-R contains 5 sub-scores, each one representing one cognitive domain: attention/orientation (18 points), memory (26 points), fluency (14 points), language (26 points), and visuospatial (16 points). ACE-R maximum score is 100, composed of the addition of all domains.

The GDS is a 15-item scale designed to measure depression in older people with dementia. One well-known advantage of GDS is the forced choice (yes/no) response format, which requires very little cognitive involvement. A score of 0 to 5 is normal. A score greater than 5 suggests depression. Descriptive analyses included absolute frequencies and percentages for categorical variables and measures of central tendency for continuous variables based on their distribution. Subsequently, intertemporal comparisons were performed using parametric or non-parametric tests. The power of the study was obtained by calculating the effect size resulting from the change in MMSE between baseline and T2 (first post-treatment follow-up). The effect size obtained, assuming a first-species error of 0.05 in a model of one-tail mean difference for paired samples, gives a statistical power of over 90%.

### 2.5. QUAL Topics and Analysis

After six months of maintenance therapy, QUAL data were collected via a semi-structured topic-guide interview that addressed two main topics: participants’ satisfaction with the therapy and their state of mind during the online rehabilitation sessions (two examples of questions addressing this topic were ‘How did you feel during the online meetings?’ and ‘What did you like and dislike about these?’), and the experience with the technology (e.g., ‘Did you connect to the online platform autonomously?’; ‘What are the pros and cons of using the technology?’; and, ‘Would you prefer face-to-face, online, or blended meetings and why?’). The interviews were digitally recorded and transcribed verbatim. Two researchers independently analyzed textual data deductively and inductively, discussed the results with a third researcher, and reached a consensus in order to minimize personal bias and interpretation errors, thereby ensuring validity and reliability [47]. The study’s trustworthiness, that is, its credibility, transferability, dependability, and confirmability [48], was ensured by tracking internal analysis processes, for instance through frequent debriefing and reflective commentary sessions within the research team [49,50]. Data were analyzed by means of the qualitative content analysis method [51] using MaxQda software (version 2020). Specific topics and content related to the study’s objectives were filtered out of the textual material. The extracted chunks of text were named by the categories and sub-categories that were developed inductively and codified using a code-tree.

## 3. Results

### 3.1. Quantitative Findings

Table 1 illustrates the respondents’ demographic profile. The mean age of participants was 73.1 (SD 4.8) years. The majority were men (61.1%) with a lower secondary education or above (SD 4.1). The mean MMSE score was 27.8 (SD 1.7) at baseline (Table 2).

There was a slight non-significant decline between T0 and T1 (*p* = 0.875) in the general cognitive functions measured using MMSE, whereas a significant increase in the general cognitive level was highlighted between T1 and T2 (*p* = 0.001) after the intensive treatment. The MMSE score remained stable between T2 and T3 (*p* = 1.000) after the maintenance therapy (Figure 2a). The ACE-R total score increased between T1 and T2 (*p* = 0.001) after the intensive treatment, and even more between T2 and T3 (*p* = 0.003) after the maintenance therapy, highlighting a continuous improvement in the performance of participants (Figure 2b). A significant increase in the attention/orientation domain was highlighted between T1 and T2 (*p* = 0.001) after the intensive treatment, and even more between T2 and T3 (*p* = 0.003) after the maintenance therapy (Figure 2c). Participants showed an initial and significant improvement in memory performance between T1 and T2 (*p* = 0.012) after the intensive treatment, whereas the score remained stable between T2 and T3 (*p* = 0.359) after the maintenance therapy (Figure 2d). In the fluency domain, there was no significant difference between T1 and T2 (*p* = 0.573), which, on the other hand, was stressed between T2 and T3 (*p* = 0.002) after about 6 months of treatment (Figure 2e). Participants showed an initial and significant improvement in language performance between T1 and T2 (*p* = 0.002) after the intensive treatment, whereas the score remained stable between T2 and T3 (*p* = 0.454) after the maintenance therapy (Figure 2f). As for memory and language scores, a significant increase in the visuospatial domain was also highlighted between T1 and T2 (*p* = 0.026) after the intensive treatment, whereas the score remained stable between T2 and T3 (*p* = 0.749) after the maintenance therapy (Figure 2g). There was a decrease in the level of depression measured through the GDS between T1 (mean 3.78, SD 2.84) and T3 (mean 2.83, SD 2.09), which was never significant (T1–T2: *p* = 0.120; T2-T3: *p* = 0.449) (Figure 2h). Quantitative longitudinal findings are also available in the Appendix A.

### 3.2. Qualitative Findings

Out of 18 PD patients enrolled in the study, 15 agreed to be interviewed and three refused due to health issues. Two of the 15 interviews conducted were found to be invalid due to respondents’ language difficulties, rendering these insufficiently informative. Table 3 provides an overview of the main codes and sub-codes identified as a result of the content analysis for each PD patient interviewed (ID).

Before detailing the reasons behind the advantages and the disadvantages of the intervention, it is worth mentioning that the majority of patients referred to it as ‘training.’ The use of this word instead of ‘treatment’ seems to be important, since it can imply that participants did not view themselves as ‘patients,’ but rather as ‘learners,’ that is, people who are not included in any kind of classification by disease. Most participants were ‘very’ or ‘very much’ appreciative of the online treatment, describing it as ‘useful’ and ‘helpful.’ Some participants liked the topics discussed during the online meetings and the healthcare professionals’ closeness; others liked the relational aspects of the treatment, specifically the ability to maintain contact with others, as this alleviated the social isolation induced by the lockdown’s physical distancing:

*I liked the ‘training’ because the weekly commitment helped me to make new friends. Without detracting from the validity of the ‘games’ we played on a psychological, knowledge, cultural, and other levels, the act of interacting with other people is the aspect that I liked the most*.(P5)

The main advantage of the treatment was the positive impact on the participants’ mood. In fact, eight patients reported feeling ‘calm’ and at peace during the online meetings, and two of them credited this to the psychologist who led the activities and moderated group discussions. Four people said they felt ‘happy’ because they had the chance to meet people they referred to as ‘friends’ and communicate thoughts and feelings. One participant drew our attention more than others in the group of ‘happy’ participants because he/she said: ‘*I was happy because there were some people I already knew on the course and seeing how well they had coped on the whole (i.e., to COVID-19 and isolation), made me happy’* (P10).

Four participants reported feeling ‘stimulated’ and ‘interested’ in the activities proposed by the group moderator. However, one person described himself as ‘bothered’ since he/she found the treatment to be ‘useless’: “*First of all, the connection was frequently unreliable, and it took a lot of time to restore everything. It’s practically useless, which is why it didn’t appeal to me […]. However, I believe that something is always preferable to nothing […]. In my opinion, however, there were more negative aspects, and many times I was almost tempted to not even go online at all”* (P12).

According to the participants, other advantages of the online treatment were the ability to continue meeting, discussing, and doing exercises despite the physical distancing measures imposed by the pandemic, and subsequently, to continue combating PD symptoms, while reducing mobility and saving time, as depicted by the following quotations:

*“Carrying out these cognitive activities is very positive. It helps us to control the discomfort associated with ‘Parkinson’s’ problems”*.(P3)

*“The advantages are time and risk savings. When one is on the street, one is increasingly at risk, and because of the time of maybe one to five minutes that it takes to connect and teach, instead of wasting time on those 45 min of travel”*.(P9)

The main disadvantage of the online treatment was the lack of physical interaction which, combined with the small group size, made the at-a-distance sessions less fun than face-to-face meetings:

*“We were better off when we were physically present. We even had a few more laughs when we were face-to-face. We had fun when the group consisted of ten people and there was harmony among us. There was a lot of laughter, even when we were online, but the group was smaller with only four people. There was a heightened sense of emotion when we were physically present in the group to carry out the activities”*.(P6)

The main difficulty of the online treatment stemmed from the lack of digital literacy of some participants who approached it with apprehension, fearful that they would not be able to solve any of the technical problems that might occur. Most respondents indeed, had difficulties accessing the platform:

*“I didn’t know how to connect the first time, but after I persisted, I managed. The others also encountered the same difficulties, and we all tried to be in contact at the same time”*.(P6)

Seven out of 13 participants received help from family members, spouses, children, and grandchildren at least for the first few times of using the PC or tablet, to access the platform, and to use it correctly, i.e., camera and microphone activation and muting. Six patients out of thirteen preferred to continue cognitive rehabilitation exclusively in person, four exclusively online, two suggested a blended system, and one was undecided. One quotation for each preference is reported below:

Face-to-face: *“I want the face-to-face meetings to continue because I had to go out in order to attend the meetings at the daycare center, and this stimulated me: I got dressed, put on make-up, took the car, and left home. That is, being physically present while participating required me to engage in a whole set of activities that stimulated me”*.(P6)

Online: *“Online is fine for me because we are all afflicted with ailments and, therefore, being able to meet and talk in such a calm manner through these new technological means is a great opportunity”*.(P5)

Blended: *“It would be nice to interact again while being physically present and safe. However, this type of exercise is also okay. Maybe it can be done in person once and then the other times here, let>’s say to meet in person”*.(P2)

Undecided: *“There are pros and cons to face-to-face or online training sessions. In presence, the group is closer, but you have to go out and you run more risks. On the other hand, you have more time online, you are calmer at home, but there is separation, there is no physical contact. I don’t know which one to choose from the two”*.(P9)

## 4. Discussion

The main novelty of this study lies in the mixed-method research approach that allowed us to bring together the quantitative and objective effects of the online cognitive rehabilitation treatment. The study shows that the online treatment had a positive impact on participants’ cognitive functions, in the medium term, as observed in previous studies [52,53], and mirrored by the QUAL finding. Since the latter indicates that the majority of PD patients in the study were interested in and stimulated by the issues presented by the group moderator (i.e., a psychologist, a professional health educator, or a healthcare professional), this suggests that online treatments have to be designed to be triggering, interesting and stimulating for older people with MCI. To this purpose, they should be co-designed with participants when they have an MCI and with their informal caregivers when they suffer from more severe cognitive impairment according to the principles of the User Centered design applied to older adults [54]. The improvement or the maintenance of the capabilities were observed immediately after the intensive treatment in all the cognitive subdomains (e.g., attention/orientation and memory) except for the fluency domain, for which a first increase was observed only between T2 and T3 (*p* = 0.002) after about 6 months of treatment. In fact, verbal fluency is a timed evaluation that requires quick access to the lexicon in the absence of external support, so it requires not only the integrity of the semantic-lexical system, but also of the executive processes. On the other hand, older people with PD typically have slowing down and deficits in executive functions including working memory, planning, and visuospatial attention, and may also need longer-lasting interventions to obtain an improvement in fluency.

The QUANT data show that the online treatment had a low impact on participants’ mood. In fact, the online intervention seems to work less effectively on an emotional level, as it lacks the most participatory component of face-to-face group sessions. Therefore, the loss of human contact with the clinician and other participants and the limited flexibility in the adoption of devices most appropriate for patients’ differing needs could hinder adherence to this intervention similar to other TR interventions [55].

However, albeit slight, such an improvement may also be the result of the slow but progressive easing of isolation measures imposed by the COVID-19 outbreak during the treatment lifetime. The QUAL analysis highlighted that the intervention improved participants’ moods by encouraging socializing. In fact, the treatment was the sole method for patients to maintain a sense of feeling part of the community during the lockdown, and some patients regarded the activation of the online intervention as a show of interest from health professionals. Not having been ‘left behind’ and forgotten contributed as much to containing cognitive and psychological decline as the rehabilitation itself. Moreover, staying connected was itself a means of alleviating anxiety in some PD patients because they could see that their friends were healthy despite the distressing news of COVID-19 infections and deaths. The intervention’s success was also contingent on the operator’s ability to put participants at ease during the use of the technological devices. Patients reported that they felt calm even when they were having problems connecting. Even if, according to the literature [56,57], only a few of the patients were fully autonomous in accessing the platform, especially the first few times, many of them acquired this ability over time. This process of familiarization with artifacts is fundamental to the positive effect of the digital intervention [58]. Therefore, the intervention was also an opportunity to improve digital health literacy. Nevertheless, in a few cases, having to ask the family caregiver to turn on the PC and/or start the internet connection made some participants feel inadequate, and others who could not ask anybody for help felt ‘alone in front of a machine. This result suggests the importance of planning some digital skills training sessions (that were unable to take place in this study due to the emergency situation). Moreover, platforms for cognitive interventions should be developed to be accessible and user-friendly; the duration and frequency of rehabilitation activities should be tailored according to patients’ characteristics; a quiet and private space, where risks of distractions and interruptions can be mitigated, is recommended; therapists should monitor adherence and performance of each session remotely during the whole period of treatment.

Furthermore, even if caregivers of the study participants were supportive and facilitated adherence to TR in daily routines, it is important to avoid their excessive involvement to limit the burden of the approach. In light of the above, future studies on TR interventions targeting older people with PD should include larger and randomized samples and should also include the perspective of their informal caregivers. Despite these challenges, it is important to consider that easy access to TR tools can produce benefits (e.g., autonomy, mood, self-efficacy, and quality of life) for PD patients with consequent positive effects on their informal caregivers as well [59].

Although the large majority of participants gave positive feedback on the online treatment, the face-to-face method remains the best option, followed by a blended approach, while the at-a-distance solution seems to represent the ‘better than nothing’ option. The opinion of PD patients in their role as end-users is in line with recent literature in the neuropsychological field. In fact, many studies consider remote communication technologies, including cognitive rehabilitation [60], as potentially effective options to support healthcare interventions, and they conceived the online sessions not to replace but rather to integrate face-to-face interventions by increasing their efficacy, accessibility, and long-term outcomes [61]. Telerehabilitation (TR) is a young telemedicine subfield that could be defined as the set of instruments and protocols aimed at providing rehabilitation at a distance and supporting both the cognitive and psychosocial needs of all patients [62]. Thus, CTR may be viewed as a valid recovery tool deriving from the reshaping of cognitive rehabilitation with the use of technologies [63]. Several studies have shown that telerehabilitation can help people with different neurodegenerative diseases [64], especially during the early stages of the disease [65]. A recent systematic review from Cotelli and colleagues [66] explored the effect of different telerehabilitation protocols on patients with mild cognitive impairment (MCI), Alzheimer’s disease (AD), and frontotemporal dementia (FTD) compared to face-to-face rehabilitation treatments. They found that cognitive telerehabilitation has comparable positive effects to classical rehabilitation in improving cognitive abilities in these neurodegenerative diseases. To our best knowledge, there are no similar studies on prodromal dementia with Lewy bodies (DLB). Therefore, the use of remote neuropsychological evaluations and treatments has been shown to lead to clinical improvements that are just as effective as conventional face-to-face interventions [67].

### Limitations

The study’s first limitation lies in the convenience and small-sized sample that does not allow for the generalization of findings. Unfortunately, it was not possible to involve more patients due to the emergency nature of the treatment, started to address the interruption of the day-care center service provision during the pandemic. For the same reason, the intervention was not designed with the participants, but every session was prepared based partly on the care professionals’ knowledge of patients’ cognitive capabilities and partly on the latter’ s suggestions collected during the treatment implementation with open questions asked during the meetings.

Other limitations are related to the QUANT or QUAL data collection methodology. As far as the QUANT data is concerned. The small sample size for this study does not allow the use of sophisticated statistical analyses. However, as can be seen from Section 3.1, the statistical significance of the variations for all the indicators was analyzed by repeated measures analysis of variance (ANOVA). This type of analysis is inferential and not merely descriptive. Moreover, multivariate analyses cannot be performed due to the small sample size, as specified in the study limitations paragraph.

Furthermore, we observed the so-called ‘ceiling effect’ during the measurement of attention/orientation and language functions [68]. This effect is a measurement limitation that occurs when the maximum or nearest-maximum attainable score on a test or measurement instrument is reached, thereby decreasing the likelihood that the testing instrument has accurately measured the intended domain. Since a large percentage of respondents’ scores are at or near the maximum possible value, it could mean that test questions were not sufficiently challenging to accurately measure respondents’ skills and/or knowledge. As regards the QUAL data, although the interview topic guide was designed to help the patients’ narrative through the adoption of a few simple and open questions, some interviews were not sufficiently informative due to both interviewees’ cognitive conditions and fluency issues and the remote at distance administration imposed by the risk of contagion from COVID-19 virus.

Due to the challenges also reported by this study, CTR is not fully integrated into the clinical practice for the treatment of older people with PD yet [69].

Additionally, in this study in-depth instruments to assess changes in the affective-emotional sphere were not used, but could be an added value to be taken into account in future studies.

## 5. Conclusions

This study had the aim of contributing to the debate on the effectiveness of the CTR by shedding light on the potential and limitations of its adoption with PD patients with mild cognitive impairment and by providing some recommendations coming from the experience. The findings showed that CTR may be effective on the cognitive functioning of PD patients, as long as the interventions are co-designed and that every activity carried out online is planned based on the participants’ needs. It is also recommended that informal caregivers are involved in the treatment from the very beginning as they are an important source of information for the patients’ profiling. The digital divide still represents a barrier to the online cognitive treatment of older adults. Thus, it is important that patients and their informal caregivers are trained in the use of digital devices and social platforms.

Further research is needed for increasing practice-driven knowledge on the effectiveness of CTR among older people suffering from Parkinson’s disease and their informal caregivers living in the community.

## Figures and Tables

**Figure 1 ijerph-19-14844-f001:**
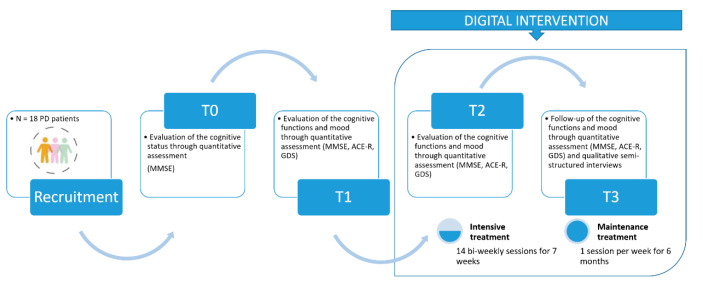
Study design overview.

**Figure 2 ijerph-19-14844-f002:**
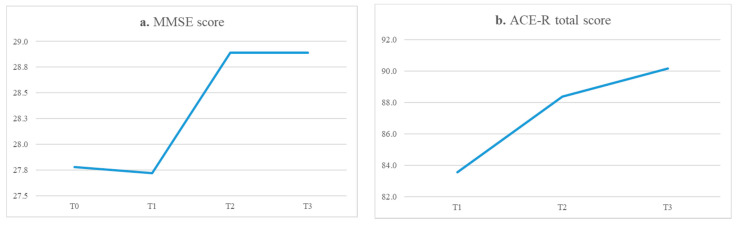
Quantitative results (Key: T0 baseline assessment, pre-COVID-19, T1 after about 8 months, without treatment, T2 after the intensive treatment, 14 twice-weekly group sessions).

**Table 1 ijerph-19-14844-t001:** Extensive neuropsychological assessment.

Neuropsychological Dimensions	Assessment Tools
Attention and executive functions	Trail Making Test A, TMT-A, and Trail Making Test, TMT-B (Amodio et al., 2002) [34]. Stroop Test (Caffarra et al., 2002) [35] Weigl’s Sorting Test (Laiacona et al., 2000) [36] Multiple Features Target Test (MFTC) time; accuracy; error (Gainotti et al., 2001) [37] Frontal Assessment Battery (FAB) (Apollonio et al., 2005) [38]
Memory	Digit span (Monaco et al., 2013) [39]
	Rey Auditory Verbal Learning Test-RAVLT, immediate and delayed recall (Carlesimo et al., 1996) [40]
	Rey-Osterrieth Complex Figure B: immediate and delayed recall (Luzzi et al., 2011) [41]
Language	Fluency for semantic categories (Costa et al., 2014) [42]
	Phonemic Fluency (FAS) (Costa et al., 2014) [42]
	Noun Naming (CAGI) (Catricalà et al., 2013) [43]
Visual constructional ability	Rey-Osterrieth Complex Figure B copy (Luzzi et al., 2011) [41]

**Table 2 ijerph-19-14844-t002:** Participants’ characteristics at baseline (T0).

Characteristics	Total
Total number of participants, n (%)	18 (100%)
Females, n (%)	7 (40%)
Males, n (%)	11 (60%)
Mean age, years	73.1 ± 4.8
Educational level	11.3 ± 4.1
MMSE	27.8 ± 1.7

**Table 3 ijerph-19-14844-t003:** Code and sub-codes of the qualitative contents’ analysis per participant.

	Code	Sub-Codes	Code	Code	Sub-Code 1	Sub-Code 2	Code
ID	Level of appreciation of the online treatment	Reasons	Mood	Experience with the technology	Advantages	Disadvantages Difficulties	Preference for the future
1	Very	Stimulating	Calm	Positive	Socialization albeit the pandemic	None	Online
2	Very	Stimulating	Calm	Positive	Socialization albeit the pandemic	None	Blended
3	Very	Stimulating	Happy	Positive	Socialization albeit the pandemic	None	In-person
4	Very much	Stimulating	Stimulated	Positive	Not to get out	Lack of physical interaction	Blended
5	Very much	A place for listening and welcoming	Stimulated	Positive	Continue the treatment albeit the pandemic	None	Online
6	Very	Stimulating	Amused	Positive	Not to get out	Lack of physical interaction	In-person
7	Very much	A place for listening and welcoming	Calm	Positive	Continue the treatment albeit the pandemic	Difficulties with technology	Online
8	Very	Stimulating	Stimulated	Positive	Not to get out	Lack of physical interaction	In-person
9	Very much	A place for listening and welcoming	Calm	Positive	Not to get out	Not to get out	Undecided
10	Very much	Stimulating	Calm	Positive	Socialization albeit the pandemic	Lack of physical interaction	Online
11	Very	Socialization	Happy Calm	Positive	Continue the treatment albeit the pandemic	Difficulties with technology	In-person
12	Not much	Waste of time: bad connection	Bothered	Positive	Not to get out	Lack of physical interaction	In-person
13	Very much	A place for listening and welcoming	Calm	Negative	Not to get out	None	In-person

## Data Availability

The data that support the findings of this study are available on request from the corresponding author. The data are not publicly available due to them containing information that could compromise research participant privacy/consent.

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
