# Peer review of "Cognitive Digital Intervention for Older Patients with Parkinson’s Disease during COVID-19: A Mixed-Method Pilot Study"

_ijerph, 2022, doi:10.3390/ijerph192214844_

Round 1

Reviewer 1 Report

First of all, I would like to thank the authors for choosing the topic of remote/telematic cognitive stimulation in the context of cognitive impairment due to Parkinson's disease. 

I will now point out aspects that raise doubts about the suitability of the article to be published in its present form: 

1) The sample is very small. We do not know some data on motor staging of the participants (UPDRS), time of evolution of the disease.... We also do not know the form of diagnosis of mild cognitive impairment (profile of this MCI, amnesic or non-amnesic, uni or multidomain...) and date of onset of cognitive complaints, the MMSE score >28 at baseline is really surprising, which could even be considered normal in a population with the level of schooling proposed. We also do not know the drugs they take, both symptomatic for Parkinson's disease and for associated comorbidities, and the possible influence of these drugs on their performance. 

2) In the introduction, Parkinson's disease is presented as a disease of elderly patients, which is not really the case. I would remove the expression of the last years of life. The usual age at diagnosis, average duration of the disease or associated life expectancy should be indicated. 

Explain better the concepts of motor and non-motor symptoms. The fluctuation of the same....

3) The tests chosen for the neuropsychological evaluation are surprising: only screening tests have been used and not the most specific ones to evaluate each domain proposed. A more complete battery and/or more screening tests should have been used for each of the domains analyzed. I think it is wrong to draw conclusions using the current tools. There is also no mention of the (limited) magnitude of the impact and only an analysis from a statistical perspective. 

On the other hand, why will the impact be assessed at the level of merely affective symptomatology? Would it not make more sense to have applied the neuropsychiatric symptom test such as the Cummings test, which better reflects the sphere of all the symptoms that may exist?

4) Conclusion: it would be necessary to compare with similar studies in other neurodegenerative diseases including prodromal Lewy if there are studies in this regard (in the introduction it is not clear what is the limit between MCI due to Parkinson's disease and the concept of prodromal Lewy), but also Alzheimer's disease. 

5) The qualitative analysis should be restructured: perceived advantages and disadvantages and difficulties in its implementation. Reduce the length, at present too long without providing much information. 

Author Response

We want to thank Rev 1 for all the precious suggestions. Here are point-by-point responses.

Reviewer 1

Comments and Suggestions for Authors

First of all, I would like to thank the authors for choosing the topic of remote/telematic cognitive stimulation in the context of cognitive impairment due to Parkinson's disease. 

I will now point out aspects that raise doubts about the suitability of the article to be published in its present form: 

  • The sample is very small. We do not know some data on motor staging of the participants (UPDRS), time of evolution of the disease.... We also do not know the form of diagnosis of mild cognitive impairment (profile of this MCI, amnesic or non-amnesic, uni or multidomain...) and date of onset of cognitive complaints, the MMSE score >28 at baseline is really surprising, which could even be considered normal in a population with the level of schooling proposed. We also do not know the drugs they take, both symptomatic for Parkinson's disease and for associated comorbidities, and the possible influence of these drugs on their performance. 

AUTHORS: The required infos were added in a new Table number 1 and the following text has been added:

“We included 12 patients with single non-amnesic type MCI, 4 with multiple non-amnesic type MCI, and 2 with multiple amnesic type MCI. The onset of cognitive symptoms was between 6 and 20 months before the diagnosis of MCI. UPDRS mean scores were: Motor 18.35±5.98; UPDRS II (ADL) 13.58 ± 4.44; Total 31.94±9.26. One patient was not receiving any PD treatment, five patients were under low-dose levodopa monotherapy, the other patients were receiving levodopa combined with other antiparkinsonian treatments (eight patients also had a prescription for dopamine agonists, while four also MAO-B inhibitors). Five patients were using antihypertensive medications, none of the participants were taking medications which may influence the cognitive evaluation”.

  • In the introduction, Parkinson's disease is presented as a disease of elderly patients, which is not really the case. I would remove the expression of the last years of life. The usual age at diagnosis, average duration of the disease or associated life expectancy should be indicated. 

AUTHORS: We rephrased the sentence, withdrew the expression “mainly affecting people in the later years of life”, we added the average age of diagnosis and specified that “PD increases with age, reaching a prevalence of 2.6% in people aged 85 to 89 years”.

  • Explain better the concepts of motor and non-motor symptoms. The fluctuation of the same....

AUTHORS: we specified moto and non-motor PD sympthoms as follows: “PD can impair motor functions (i.e. bradykinesia, rest tremor, and rigidity, ascribed to the loss of dopaminergic neurons, and those involving posture, balance, and gait), as well as non-motor functions i.e., sleep disorders, a spectrum of neuropsychiatric symptoms (e.g. depression, apathy, hallucinations), autonomic disorders (e.g. constipation and urinary disturbances), and cognitive impairment (e.g. involvement of executive functions, memory, and visuospatial functions) up to dementia [3]”.

  • The tests chosen for the neuropsychological evaluation are surprising: only screening tests have been used and not the most specific ones to evaluate each domain proposed. A more complete battery and/or more screening tests should have been used for each of the domains analyzed. I think it is wrong to draw conclusions using the current tools. There is also no mention of the (limited) magnitude of the impact and only an analysis from a statistical perspective. 

AUTHORS: The diagnosis of MCI was made according to the Litvan’s criteria and a face to face extensive neuropsychological battery was administred, shortly before the outbreack of pandemic. The battery has been reported in the text. It was not possible to administrer most of these tests remotly. Consequently the four waves assessment was performed using screening tools such as MMSE and ACE-R, because they were better suited to remote administration (reported in the text).

See:

- Cognitive testing in the COVID-19 era: can existing screeners be adapted for telephone use? Andrew J Larner. Neurodegener Dis Manag. 2020 Oct : 10.2217/nmt-2020-0040.

- Feasibility of remote Memory Clinics using the plan, do, study, act (PDSA) cycle. Jemima T Collins, Biju Mohamed. Age Ageing . 2021 Nov 10;50(6):2259-2263.

- Telemedicine and the mini-mental state examination: assessment from a distance. Elizabeth L Ciemins, Barbara Holloway, Patricia Jay Coon, Thelma McClosky-Armstrong, Sung-Joon Min. Telemed J E Health. 2009 Jun;15(5):476-8.

5)  On the other hand, why will the impact be assessed at the level of merely affective symptomatology? Would it not make more sense to have applied the neuropsychiatric symptom test such as the Cummings test, which better reflects the sphere of all the symptoms that may exist?

AUTHORS: The Cummings Neuropsychiatric Inventory is a dementia-specific tool and in our opinion it could not be suitable for MCI patients who had no relevant behavioural disorders.

  • Conclusion: it would be necessary to compare with similar studies in other neurodegenerative diseases including prodromal Lewy if there are studies in this regard (in the introduction it is not clear what is the limit between MCI due to Parkinson's disease and the concept of prodromal Lewy), but also Alzheimer's disease. 

AUTHORS: The request of comparing the result with similar studies has been addressed in the manuscript body.

  • The qualitative analysis should be restructured: perceived advantages and disadvantages and difficulties in its implementation. Reduce the length, at present too long without providing much information. 

AUTHORS: The QUAL results were restructured by highlighting perceived advantages and disadvantages of the online treatment, and the text was shortened as much as possible.

Reviewer 2 Report

The work sent for review is interesting and concerns a very important aspect - Cognitive digital intervention for older patients with Parkinson's disease during COVID-19.
However, the manuscript contains several problems and doubts:
introduction
Aim - has not been clearly specified. Needs improvement and clear definition. The proposed by the authors is more a description of the end points.
Material and method:
A very small group of patients was incluted in the study.
Have patients had a comprehensive geriatric evaluation. What was the age of the patients included in the study, were they elderly? It is worth including this information in the study inclusion criteria.
Results: There is no reference to table 2 in the text, which is illegible. The results are descriptive, no statistical calculations were used, which makes them unreliable. In my opinion, this is a significant disqualifying point for the job.
Discussion: in discus, it is worth not to repeat the results. We refer to our results in comparison to other researchers or translate our results.
Applications should not contain references to the literature. These are our conclusions that result from the results obtained.

Author Response

We want to thank Rev 2 for all the precious suggestions. Here are point-by-point responses.

Reviewer 2

Comments and Suggestions for Authors

The work sent for review is interesting and concerns a very important aspect - Cognitive digital intervention for older patients with Parkinson's disease during COVID-19.
However, the manuscript contains several problems and doubts:

  • introduction
    Aim - has not been clearly specified. Needs improvement and clear definition. The proposed by the authors is more a description of the end points.

AUTHORS: The aim of the study was re-written clearly.

  • Material and method:
    A very small group of patients was incluted in the study.

AUTHORS: The limitation of the small sample size was explained better in the limitation paragraph.

3) Have patients had a comprehensive geriatric evaluation. What was the age of the patients included in the study, were they elderly? It is worth including this information in the study inclusion criteria.

AUTHORS: The age of the patients included in the study was added to the manuscript body (paragraph 2.3)

  • Results: There is no reference to table 2 in the text, which is illegible.

AUTHORS: Reference to Table 2 (Now Table 3) was added.

The results are descriptive, no statistical calculations were used, which makes them unreliable. In my opinion, this is a significant disqualifying point for the job.

AUTHORS: To address this point the following text was added to the manuscript body (Par. 4.1): “The small sample size for this study does not allow the use of sophisticated statistical analyses. However, as can be seen from paragraph 3.1, the statistical significance of the varia-tions for all the indicators was analysed by repeated measures analysis of variance (ANOVA). This type of analysis is inferential and not merely descriptive. Moreover, multivariate analyses cannot be performed due to the small sample size, as specified in the study limitations paragraph”.

4) Discussion: in discus, it is worth not to repeat the results. We refer to our results in comparison to other researchers or translate our results.
Applications should not contain references to the literature. These are our conclusions that result from the results obtained.

AUTHORS: All repetitions of results have been deleted and the results were compared to previous similar studies. Every reference was moved away from the Conclusions and the session was mostly re-written.

Round 2

Reviewer 1 Report

The authors have attempted to remedy most of the drawbacks previously raised. Although I still perceive methodological limitations, I believe that they have substantially improved the text. However, I disagree with them on the reason for not using the Cummings Neuropsychiatric Symptom Inventory (NPI). Although it was initially used in the context of dementia, there are many papers that support its use in the MCI stage, and I believe it would have added value. 

Author Response

Dear Reviewer, thank you for this last suggestion. We included the following sentences at the end of the discussion where the limitations of the study are reported: "Additionally, in this study in-depth instruments to assess changes on the affective-emotional sphere were not used, but could be an added value to be taken into account in future studies."